# HIV co-infection is associated with reduced *Mycobacterium tuberculosis* transmissibility in sub-Saharan Africa

Etthel M. Windels[1,2,3]*, Eddie M. Wampande[4], Moses L. Joloba[4], W. Henry Boom[5], Galo A. Goig[2,6], Helen Cox[7], Jerry Hella[8], Sonia Borrell[2,6], Sebastien Gagneux[2,6�], Daniela Brites[2,6�], Tanja Stadler[1,3�]*

**1** Department of Biosystems Science and Engineering, ETH Zürich, Basel, Switzerland, **2** Swiss Tropical and Public Health Institute, Allschwil, Switzerland, **3** Swiss Institute of Bioinformatics, Lausanne, Switzerland, **4** Makerere University, Kampala, Uganda, **5** Case Western Reserve University and University Hospitals Cleveland Medical Center, Cleveland, Ohio, United States of America, **6** University of Basel, Basel, Switzerland, **7** University of Cape Town, Cape Town, South Africa, **8** Ifakara Health Institute, Dar es Salaam, Tanzania

460 These authors contributed equally to this work as senior authors.
* etthel.windels@bsse.ethz.ch (EMW); tanja.stadler@bsse.ethz.ch (TS)

**Data Availability Statement:** 185 genome sequences collected in Kampala, Uganda, were deposited to the European Nucleotide Archive

## Abstract

Persons living with HIV are known to be at increased risk of developing tuberculosis (TB) disease upon infection with *Mycobacterium tuberculosis* (*Mtb*). However, it has remained unclear how HIV co-infection affects subsequent *Mtb* transmission from these patients. Here, we customized a Bayesian phylodynamic framework to estimate the effects of HIV co-infection on the *Mtb* transmission dynamics from sequence data. We applied our model to four *Mtb* genomic datasets collected in sub-Saharan African countries with a generalized HIV epidemic. Our results confirm that HIV co-infection is a strong risk factor for developing active TB. Additionally, we demonstrate that HIV co-infection is associated with a reduced effective reproductive number for TB. Stratifying the population by CD4+ T-cell count yielded similar results, suggesting that, in this context, CD4+ T-cell count is not a better predictor of *Mtb* transmissibility than HIV infection status alone. Together, our genome-based analyses complement observational household contact studies, and more firmly establish the negative association between HIV co-infection and *Mtb* transmissibility.

## Author summary

Many sub-Saharan African countries have seen a considerable rise in TB incidence since the introduction of HIV, suggesting a strong interaction between HIV and TB epidemics. HIV infection is recognized as an important risk factor for developing TB, but the contribution of HIV-infected TB cases to further *Mtb* transmission is poorly understood. In this study, we analyzed four sets of *Mtb* genomic sequences collected in different countries, comprising sequences from HIV-negative and HIV-positive TB patients. We applied a phylodynamic model to these sequences, aimed at inferring transmission dynamics within

(ENA) at EBI, registered under project accession numbers PRJEB11460 (https://www.ebi.ac.uk/ena/browser/view/PRJEB11460), PRJNA354716 (https://www.ebi.ac.uk/ena/browser/view/PRJNA354716), and PRJEB64921 (https://www.ebi.ac.uk/ena/browser/view/PRJEB64921). The code for the phylodynamic analyses, including BEAST2 XML files, is available at https://github.com/EtthelWindels/tb_hiv.

**Funding:** E.M.W. and T.S. received funding from ETH Zürich and from the European Research Council (ERC) under the European Union's Horizon 2020 research and innovation programme grant agreement No. 101001077 (PhyCogy). This study was further supported by grants from ERC (883582-ECOEVODRTB) and the Swiss National Science Foundation (310030_188888, 320030-227432, and CRSII5_213514) to S.G., and a Swiss and South Africa joint research award to H.C. and S.G. (IZLSZ3_170834). The funders did not play any role in the study design, data collection and analysis, decision to publish, or preparation of the manuscript.

**Competing interests:** The authors have declared that no competing interests exist.

and between different host populations. While our findings support that HIV is a strong risk factor for TB, we show that HIV-positive TB cases generate a significantly lower number of secondary TB cases than HIV-negative cases. This suggests that HIV-positive cases mainly act as sinks in *Mtb* transmission chains, while HIV-negative cases are a major source of transmission.

## Introduction

The human immunodeficiency virus 1 (HIV) was first introduced into the human population in the beginning of the 20th century through a zoonotic transmission event [1, 2]. Its silent spread in the following decades resulted in a globally established HIV epidemic, disproportionally affecting sub-Saharan Africa [3]. In addition to directly related healthcare challenges, the high prevalence of HIV in these countries has contributed to a strong rise in tuberculosis (TB) incidence rates [4–9]. Accordingly, HIV co-infection in *Mycobacterium tuberculosis* (*Mtb*)-infected patients has been associated with an increased risk of progression to active TB disease, an increased risk of recurrent *Mtb* infection, and an increased TB case-fatality rate [5, 9–13]. Despite our incomplete understanding of the interactions between TB, HIV, and the human immune system, it is widely accepted that the depletion of CD4+ T-cells underlies the high TB susceptibility and mortality in HIV-infected individuals [9, 11, 14–16].

While many studies support this increased susceptibility to TB disease, the effects of HIV co-infection on the generation of secondary TB cases remain poorly understood [9, 17]. The HIV-associated reduction in CD4+ T-cell count has been shown to be associated with an altered TB disease presentation, including lower levels of lung cavitation, lower bacterial loads in the sputum, and a higher likelihood of extrapulmonary TB [9, 18]. This distinct lung pathology could result in reduced *Mtb* transmission, as transmission is mainly driven by the formation of aerosols from infected lungs, and lung cavitations are known to enhance transmission [19, 20].

Several household contact studies have indicated reduced infectiousness of HIV-positive TB index cases [21–26], although it should be noted that many of these studies only considered sputum smear-negative HIV patients or patients with considerably reduced CD4+ T-cell counts [23–26]. In contrast, a meta-analysis [27] and a more recent whole-genome sequence analysis of multidrug-resistant (MDR) *Mtb* isolates [28] found no association between HIV co-infection and the probability of *Mtb* transmission, although this could result from the low sensitivity of the methodology used [29].

While the altered lung pathology of HIV co-infected TB cases could affect the rate at which these individuals transmit *Mtb*, the number of secondary cases generated is also determined by how long these individuals remain infectious for *Mtb*. Previous studies on the duration of *Mtb* infectiousness in HIV-positive patients showed mixed results. Several studies indicate that HIV co-infection shortens the *Mtb* infectious period [30–32]. This could be explained by a higher TB mortality rate or faster TB disease progression, resulting in a more timely diagnosis and initiation of treatment [8, 9, 13, 33, 34]. In contrast, one study estimated that the time until accessing TB treatment was longer for HIV-positive patients [35], which could be explained by a postponed diagnosis due to barriers to care for HIV patients, or by higher rates of smear-negativity and an atypical disease presentation.

As the effects of HIV co-infection on the *Mtb* transmission rate and infectious period are not well established, it remains unclear how HIV influences the overall transmissibility of *Mtb*. Here we analyzed four *Mtb* genomic datasets from sub-Saharan African countries with a high

burden of HIV (Malawi, South Africa, Tanzania, and Uganda). In particular, we applied a Bayesian phylodynamic model, coupling an epidemiological model with a model of sequence evolution, to investigate how HIV co-infection affects the *Mtb* transmission dynamics. Our phylodynamic model stratifies the TB patient population by HIV infection status, and is parametrized with the aim of estimating the effect of HIV co-infection on both the risk of developing active TB upon exposure and the average number of secondary TB cases generated per infected individual (i.e., the effective reproductive number). Our results confirm that HIV co-infection is associated with an increased risk of developing active TB, and at the same time provide evidence for reduced *Mtb* transmissibility from HIV-positive TB cases.

## Results

We analyzed complete *Mtb* genomes collected from TB patients in four sub-Saharan African countries: 1,209 sequences from Karonga District, Malawi (1995–2011) [36], 1,133 sequences from Khayelitsha, Cape Town, South Africa (2008–2018) [37], 1,074 sequences from Temeke District, Dar es Salaam, Tanzania (2013–2019) [38], and 185 sequences from Kampala, Uganda (1995–2012) [39, 40] (see Materials and methods for details on the study populations). The sequences from Uganda have been used partially in other studies [41, 42] but are analyzed here together for the first time. *Mtb* lineage distributions per sampling location are shown in S1 Table.

To quantify the effects of HIV co-infection on *Mtb* transmission in these locations, we customized a phylodynamic model based on the structured birth-death model [43], with the TB patient population stratified by HIV infection status as determined at the time of TB diagnosis (S1 Fig). In this model, transmission of *Mtb* within and between subpopulations (i.e., HIV-negative and HIV-positive TB cases) is described with different transmission rates (number of transmission events per infected individual per unit of time), and each subpopulation is additionally characterized by a rate of becoming uninfectious (1/infectious period) and a sampling rate. The ratio of the transmission rate and the becoming uninfectious rate corresponds to the effective reproductive number ($R_e$), representing the average number of secondary TB cases that one infected individual generates in the same or the other subpopulation. To explicitly model the effects of HIV, we reparametrized this model with (1) a base $R_e$, corresponding to the $R_e$ for TB in a purely HIV-negative population ($R_e^b$), (2) a parameter for the multiplicative effect of HIV co-infection on the $R_e$ for TB, at the donor side of transmission ($f_1$), (3) a parameter for the multiplicative effect of HIV co-infection on the risk of getting diagnosed for active TB disease after contact with an infectious individual ($f_2$), (4) the rate at which HIV-negative individuals become *Mtb* uninfectious ($\delta^-$), (5) a parameter for the multiplicative effect of HIV co-infection on the rate of becoming *Mtb* uninfectious ($f_3$), and (6) a parameter for the HIV prevalence in the general population ($p_{HIV}$) (S1 Fig; see Materials and methods for more details on the phylodynamic model). The HIV prevalence was included to account for the HIV-negative and HIV-positive susceptible population sizes, which in turn influence the contact probabilities and thus the $R_e$ in each location. HIV prevalences were not estimated from the genomic data, but set to time-varying levels based on location-specific prevalence data from World Bank [44–47] (S2 Fig). To improve the identifiability of the parameters of interest, the becoming uninfectious rate for HIV-negative individuals was fixed to 1 year$^{-1}$. This corresponds to an average *Mtb* infectious period of 1 year, which is within the range of previous estimates [30–32, 35]. We assumed no migration in the model, implying that the HIV status does not change during the course of an *Mtb* infection. We also assumed a constant $R_e^b$, as justified by the results of an unstructured birth-death skyline analysis [48] (S3 Fig). Each sampling location was analyzed independently. For each location, we inferred *Mtb* lineage-specific

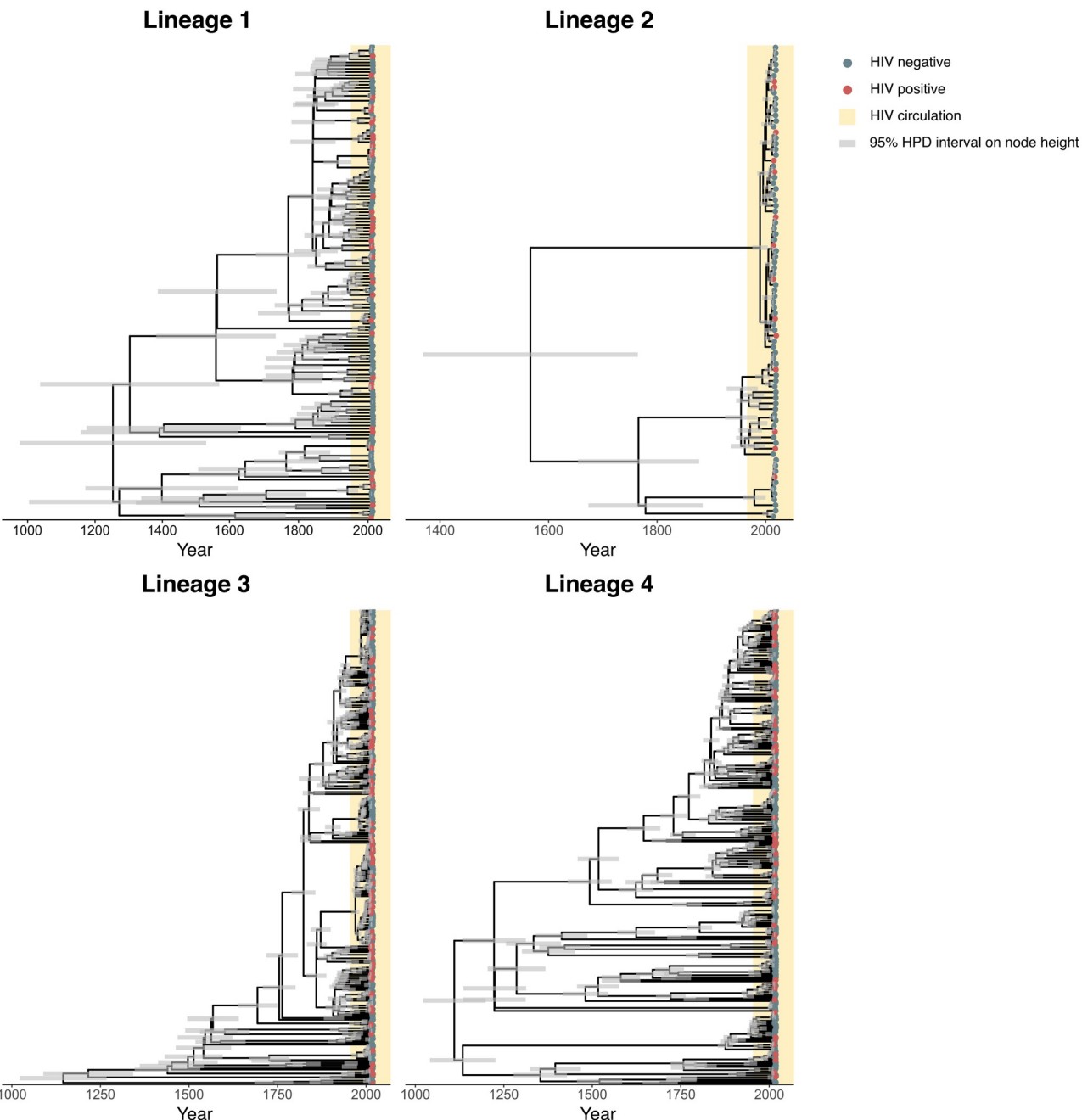

**Fig 1. Posterior maximum clade credibility trees of *Mtb* isolates from Tanzania.** Posterior maximum clade credibility trees per lineage, summarizing the posterior tree distribution resulting from the phylodynamic analyses on the *Mtb* sequences from Tanzania, with tips labeled by HIV infection status. Trees of isolates from the other locations are shown in S4–S6 Figs.

phylogenetic trees, with each tree modelled as having an independent origin and evolutionary parameters. Epidemiological parameters were assumed to be the same for all lineages co-circulating within a given location. All parameters were estimated with the Bayesian phylogenetics package BEAST2 [49, 50], with prior distributions summarized in S2 Table.

The posterior maximum clade credibility trees of *Mtb* isolates from Tanzania are displayed in Fig 1 (trees for the other locations are shown in S4–S6 Figs), indicating limited clustering of

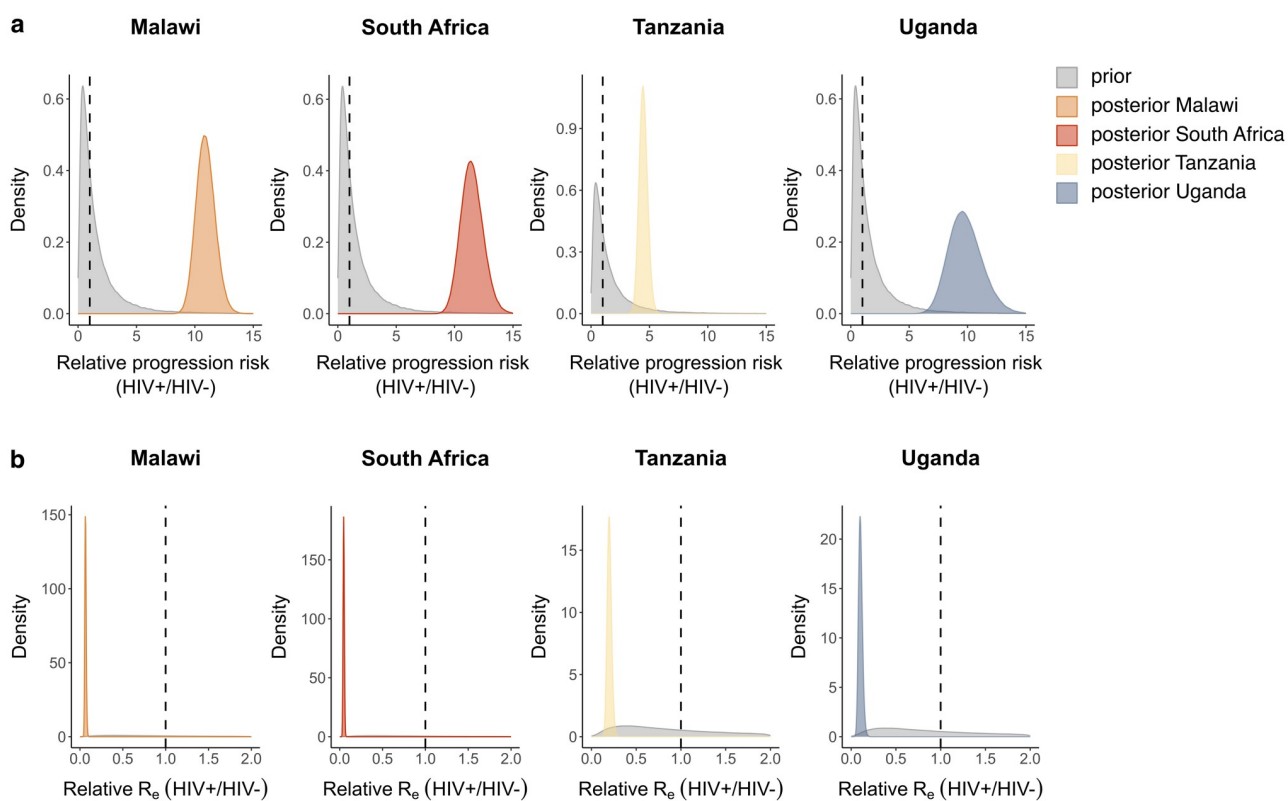

**Fig 2. Phylodynamic estimates of the effects of HIV co-infection on *Mtb* transmission.** Prior (grey) and posterior (coloured) distributions per sampling location of the estimates for a) the relative risk of developing active TB upon exposure (HIV-positive relative to HIV-negative individuals), b) the relative $R_e$ for TB. For all posterior distributions, the 95% HPD intervals do not contain 1.

HIV-positive patients. The posterior distributions of the parameter estimates show that the TB disease development risk for HIV-positive relative to HIV-negative individuals is significantly higher than 1 in each sampled location, with posterior means ranging from 4.48 to 11.49 (Fig 2A). These results suggest that HIV-positive individuals have a 4- to 11-fold increased risk of developing active TB upon exposure, which is in accordance with the increased TB incidence rate observed in HIV patients [9, 10, 12, 13], and can thus be considered as a 'positive control' in our analyses. Posterior estimates for the relative $R_e$ (HIV-positive relative to HIV-negative individuals) are all significantly lower than 1, with posterior means ranging from 0.047 to 0.20, implying that HIV co-infection is associated with a 5- to 21-fold reduction in $R_e$ for TB (Fig 2B). This could result from an altered TB disease presentation, as supported by the significantly lower chest X-ray scores of HIV-positive compared to HIV-negative TB patients in Uganda (Welch's two-sample *t*-test, $p = 0.044$). Similarly, HIV-positive TB patients from Tanzania showed reduced chest X-ray scores (Welch's two-sample *t*-test, $p < 0.001$), less cavity development ($\chi^2$-test, $p < 0.001$), and lower bacterial loads in the sputum (Welch's two-sample *t*-test, $p = 0.0027$) [38], reflecting a distinct lung pathology, consistent with reduced infectiousness. Furthermore, clinical data from Malawi and South Africa showed that HIV-positive patients were strongly associated with extrapulmonary TB ($\chi^2$-test, $p < 0.001$ for both datasets), which is non-transmissible. All these observations are consistent with previous studies [9, 18].

We could not identify a clear effect of HIV co-infection on the *Mtb* infectious period, due to wide posterior distributions for some locations and conflicting results for others (S7 Fig). However, irrespective of the infectious period, HIV co-infection was associated with a reduced *Mtb* transmission rate (S7 Fig), suggesting that HIV mainly affects the strength, rather than the duration of infectiousness.

To investigate how much our results are impacted by the assumptions on the becoming uninfectious rate of HIV-negative individuals, we repeated the analyses with fixed values of 0.5 year$^{-1}$ and 2 year$^{-1}$ for this parameter (S8 Fig). Further sensitivity analyses comprised different priors on the parameters for the HIV effects on *Mtb* transmission and TB disease development risk (S9 Fig), fixing the clock rate to either $10^{-8}$ or $10^{-7}$ substitutions per site per year (S10 Fig; see S3 Table for the clock rate estimates resulting from the main analyses), setting the HIV prevalence to local instead of national estimates when available (S11 Fig; see Materials and methods), and assuming a twofold undersampling (e.g., due to decreased culturability) or oversampling (e.g., due to increased progression risk and better access to diagnosis) of HIV-positive compared to HIV-negative TB cases (S12 Fig). While the absolute values of the posterior estimates of interest were weakly dependent on the prior assumptions, all sensitivity analyses resulted in the same qualitative conclusions regarding the relative progression risk and relative $R_e$.

Phylodynamic birth-death estimates are not only informed by the genomic data, but also by the distribution of sample collection dates. An additional set of analyses where the sequences were ignored showed that estimates of the relative progression risk with and without the genomic data were in close agreement, with only slightly shifted posterior distributions and more certainty in the estimates when the sequences were included (S13 Fig), suggesting that they contain little information about this parameter. For the relative $R_e$, including the sequences resulted in shifted and narrower posterior distributions (S13 Fig), indicating that while the isolation dates and HIV infection status of patients are the major source of information, the genomic data further inform this parameter.

To identify potential biases introduced through model assumptions and priors, we repeated the analyses on datasets where the HIV status of the patients was permuted. These datasets still contained signal for the relative progression risk and relative $R_e$ (S14 Fig). This can be explained by the HIV prevalence being 4 to 7 times higher in the sampled TB patients than in the general population in the countries under study. Indeed, randomly assigning the HIV status using the average HIV frequency in the general population during the sampling period resulted in posterior distributions for the relative progression risk and relative $R_e$ that overlap with 1, implying no effect of HIV co-infection (S14 Fig). Together, these results demonstrate that the signal for the HIV effect parameters originates from the data rather than from model assumptions, with the parameter inference presumably being driven by 1) the high prevalence of HIV within the population of TB patients, and 2) the sampling dates and sequences informing the overall $R_e$ estimate, which in turn constrains the HIV effect parameters.

HIV patients might show different levels of CD4+ T-cell depletion, depending on the stage of HIV infection and whether the patient is on antiretroviral therapy (ART). Several studies have indicated that decreased CD4+ T-cell counts are associated with a reduced frequency of lung cavitations (see [9] for an overview), and we found a similar association in HIV patients from Uganda (Welch's two-sample *t*-test, $p = 0.030$), suggesting that CD4+ T-cell counts might be a better predictor of *Mtb* transmissibility and TB progression than the HIV infection status alone. In contrast, one study showed that TB incidence rates were increased even in HIV patients with high CD4+ T-cell counts [11], suggesting that other aspects of HIV infection might also play a role. To investigate the contribution of CD4+ T-cell counts to the observed effects of HIV co-infection on *Mtb* transmission, we repeated our phylodynamic analyses on

sequences from South Africa and Uganda, with subpopulations defined by CD4+ T-cell counts (lower resp. higher than 350 cells/$\mu$l, the threshold recommended by WHO to prioritize patients for ART) instead of HIV status (S15 Fig). For the other sampling locations, CD4+ T-cell counts were not available. These analyses resulted in similar posterior means and HPD intervals for the parameters of interest (Fig 3), suggesting that the CD4+ T-cell count can be used as a predictor of *Mtb* transmission, but that, within the context of our analyses, it is not more informative than the HIV status.

## Discussion

The effects of HIV co-infection on *Mtb* transmission have remained elusive, with previous studies yielding contradictory results. Here we used a phylodynamic approach to address the question through *Mtb* sequences and HIV/*Mtb* co-infection data sampled in four different African countries. Our phylodynamic analyses support previous studies showing that HIV/*Mtb* co-infected individuals are at high risk of developing active TB disease compared to HIV-negative *Mtb*-infected individuals. This observation serves as a 'positive control' that supports the validity our method. Moreover, we found that HIV-positive TB cases on average cause significantly fewer secondary TB cases compared to HIV-negative TB cases. These findings were reproduced independently across all four countries included in our analysis.

The finding from this and previous studies that HIV co-infection is a strong risk factor for developing active TB disease upon exposure explains why many TB epidemics in sub-Saharan Africa seem to be driven by the high HIV prevalence [4–9]. The underlying cause of this increased susceptibility to TB might be the depletion of CD4+ T-cells in HIV patients [9, 11, 14, 15]. As we could only investigate the overall risk of developing active TB disease after contact with an infectious individual, it remains unclear, based on our data, whether HIV also affects the risk of *Mtb* infection.

The consistently reduced TB $R_e$ from HIV-positive individuals, observed in all countries under study, seems to be linked to a reduced number of transmission events per infected individual per unit of time. These findings are in accordance with the reduced *Mtb* infectiousness of HIV patients previously observed in various household contact studies [21–26]. A reduced infectiousness of HIV-positive TB cases can potentially be explained by an altered TB disease presentation in HIV-positive individuals, which could in turn result from an impaired immune system. This notion is supported by significant associations between HIV infection status and clinical variables related to lung damage and bacterial burden, observed in this and previous studies [9, 18].

The TB $R_e$ of HIV-positive individuals might be additionally reduced through a shorter infectious period, due to more rapid disease progression and/or an increased mortality rate [8, 9, 13, 33, 34]. In contrast, increased bacterial drug resistance, delayed diagnosis due to an atypical disease presentation, and barriers to care for HIV patients could increase the *Mtb* infectious period of HIV-positive TB cases [35]. While we could not identify a consistent HIV effect on the infectious period, we showed that the reduced $R_e$ of HIV-positive TB cases was linked to a lower transmission rate, irrespective of the duration of the infectious period.

The effects of HIV co-infection might be complicated by ART, which alleviates the CD4+ T-cell depletion in HIV patients [51, 52]. As information on ART was lacking for most patients, we could not directly take this into account in our analyses. However, we assumed that the CD4+ T-cell measurements from the patients in South Africa and Uganda would reflect differences in ART. As the CD4+ T-cell count classification (low/high) for these patients largely overlapped with their HIV infection status (S15 Fig), the observed effects of low CD4+ T-cell counts were largely similar to the effects of HIV co-infection. ART only recently

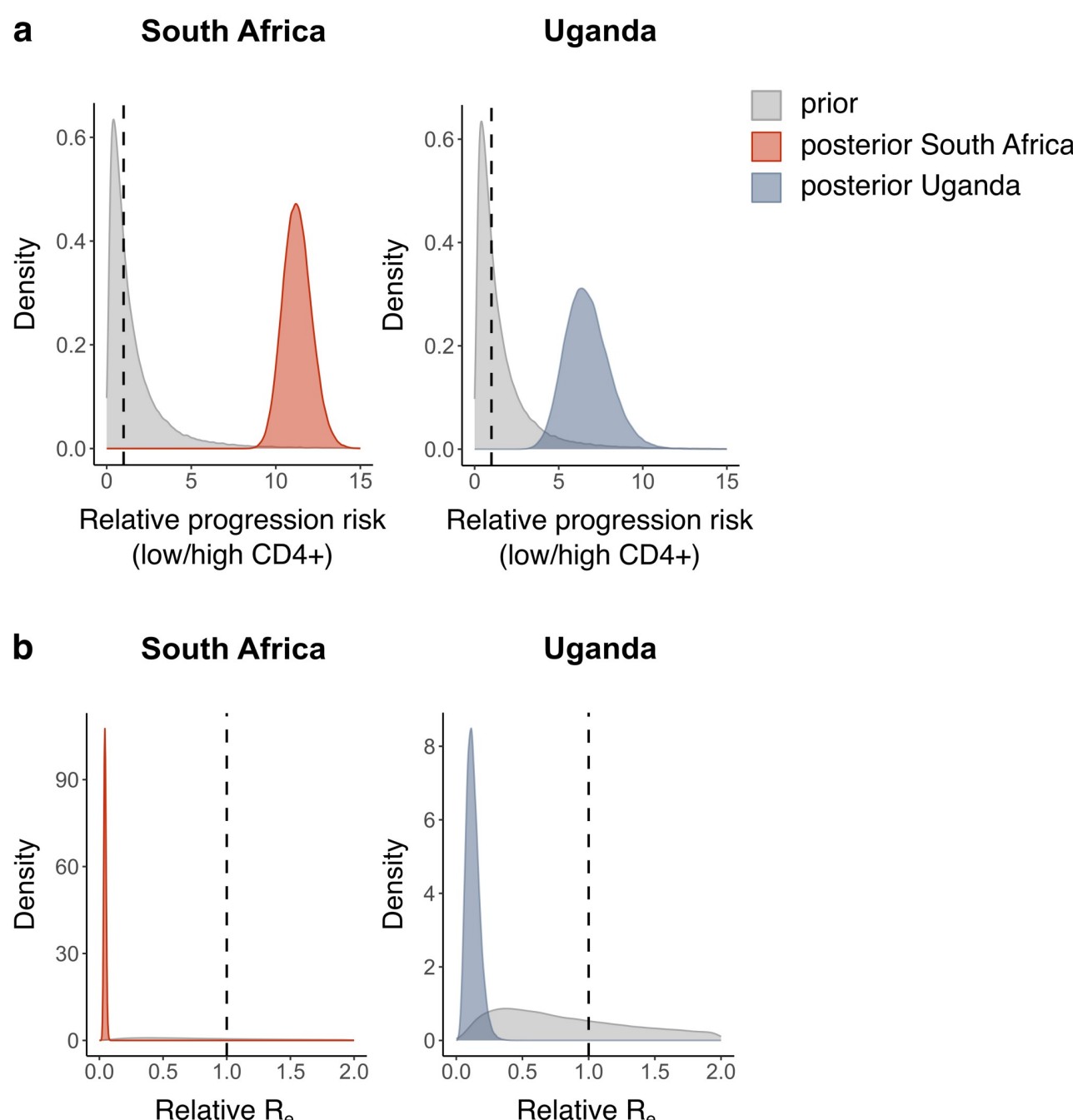

**Fig 3. CD4+ T-cell count as predictor of *Mtb* transmissibility and TB disease progression.** Prior (grey) and posterior (coloured) distributions per sampling location of the estimates for a) the relative risk of developing active TB upon exposure (individuals with low CD4+ T-cell count relative to individuals with high CD4+ T-cell count), b) the relative $R_e$ for TB. A threshold of 350 CD4+ T-cells/$\mu$l was used to classify patients. For all posterior distributions, the 95% HPD intervals do not contain 1.

became widely accessible to HIV patients in South Africa and Uganda [53, 54], which might explain why CD4+ T-cell count and HIV status are largely redundant. In other settings, ART might play a more important role in determining CD4+ T-cell counts, and consequently, CD4 + T-cell counts might be a better predictor of *Mtb* transmissibility.

HIV co-infection has been shown to be associated with rifampicin-resistant TB, potentially due to increased resistance acquisition during TB treatment [55]. While the prevalence of drug resistance during the sampling period was low in Malawi, Tanzania, and Uganda, the South-African dataset consists of rifampicin-resistant *Mtb* isolates only, indicating that HIV is associated with reduced *Mtb* transmissibility irrespective of drug resistance. Other potential confounders are poverty-related risk factors, patient sex, and patient age. However, with our current approach, these confounders are challenging to control for due to the rapidly increasing model complexity.

Notably, the major source of information on HIV effects in the model are the sampling dates and the HIV status of the TB patients, indicating that our customized birth-death model would have been able to capture most of the signal in the data even in the absence of *Mtb* sequences. A potential explanation for this limited signal in the sequence data is the fact that HIV status is a host-related factor that is not associated to the bacterial genetic background. Consequently, the *Mtb* genomes of HIV-positive TB patients are dispersed across the phylogenetic tree, resulting in high uncertainty on ancestral states and thus few informative branching events.

One limitation of our approach is that we did not account for multiple *Mtb* introductions into the study populations, nor for changes in the TB $R_e$ over time. However, HIV was most likely introduced only after the establishment of different *Mtb* lineages, and the majority of branching events informing the epidemiological parameters occur after the introduction of HIV, suggesting minimal impact on the estimates. In support of this notion, current evidence indicates that the main *Mtb* lineages circulating in these parts of Africa were introduced several centuries ago [38, 41, 56–59]. Moreover, no biases were observed when removing all information about HIV status (including prevalence) through randomization of the datasets. A second limitation is that our estimates might be biased due to an underestimation of the HIV prevalence in the general population. As this potential bias originates from the input data rather than the model, it cannot be identified with our randomization approach. A third limitation is the assumption that the probabilities of contact between and within HIV-positive and HIV-negative subpopulations are solely a function of the size of each subpopulation, ignoring any preferential contacts due to social effects. Finally, a fourth limitation of the model is the assumption that individuals infected with *Mtb* immediately become infectious (i.e., no period of latent infection). Although this assumption could affect the interpretation of the transmission rate and infectious period, the relative $R_e$ is expected to be robust, even if the duration of the latent period would be associated with HIV infection. Notably, our model does not distinguish between *Mtb*-uninfected individuals and individuals who are infected but never develop active TB disease.

Taken together, our results demonstrate that a high HIV prevalence can fuel a TB epidemic by increasing the risk for TB disease progression in HIV-positive individuals, but that these individuals do not proportionally contribute to further *Mtb* transmission. HIV-positive TB cases can thus be considered as 'sinks' in transmission chains. By contrast, HIV-negative TB cases serve as the 'sources', being disproportionally responsible for *Mtb* transmission. Our findings have implications for TB control, and call for a particular attention to HIV-negative TB cases, ideally through active case finding, thereby ensuring that these cases are diagnosed and treated as early as possible to prevent further spread of the disease.

## Materials and methods

### Ethics statement

For the study in Uganda, the institutional review boards at University Hospitals of Cleveland in the United States and AIDS Research Council in Uganda reviewed the study protocols and final approval was obtained from the Uganda National Council for Science and Technology. Written informed consent was obtained from all patients that participated in the study. All participants were given appropriate pre- and post-test HIV counseling and AIDS education. The protocols and the procedures for the protection of human subjects were approved by the Uganda National Council Ethics Committee and the Institutional Ethics Review Board at Makerere University, Kampala.

### Study populations

All datasets in this study consist of whole genome sequences (Illumina) of *Mtb* strains collected in countries with a generalized HIV epidemic. All TB cases were identified through passive case finding.

**Malawi.** Raw Illumina reads were retrieved from the European Nucleotide Archive (project accession numbers PRJEB2358 and PRJEB2794). These sequences were obtained from adults with culture-confirmed pulmonary or extrapulmonary TB diagnosed at the hospital and peripheral health care centres in Karonga District, northern Malawi between 1995 and 2011 [36]. Information about the HIV status of the patients was kindly provided by the authors of the study ($n = 1,209$). The incidence of smear-positive TB in adults in the district during the sampling period corresponds to 87–124 cases per 100,000 people per year [36].

**South Africa.** We used previously sequenced isolates from a retrospective cohort study of individuals routinely diagnosed with rifampicin-resistant (RR) or multidrug-resistant (MDR) pulmonary or extrapulmonary TB in Khayelitsha, Cape Town, South Africa between 2008 and 2018 (raw reads available in the European Nucleotide Archive under project accession numbers PRJEB45389 and PRJNA670836) [37]. Information about the HIV status was available for 1,133/1,162 patients and CD4+ T-cell counts were available for 1,110/1,162 patients. The TB notification rate in Khayelitsha was estimated around 80 RR/MDR cases per 100,000 people per year [55].

**Tanzania.** We used data from the National TB and Leprosy Programme—Tanzania, including previously sequenced isolates from a cohort of sputum smear-positive and GeneXpert-positive adult pulmonary TB patients prospectively recruited at the Temeke District hospital in Dar es Salaam, Tanzania between 2013 and 2019 (raw reads available in the European Nucleotide Archive under project accession number PRJEB49562) [38]. Information about the HIV status was available for 1,074/1,082 patients. In 2020, 3,994 TB cases were notified in Temeke (Jerry Hella, personal communication).

**Uganda.** Bacterial isolates and clinical data were obtained from pulmonary TB patients recruited in two large household contact studies. An initial study was conducted from 1995 to 1999 to describe the epidemiology of TB in urban Kampala, Uganda [39, 60]. The second is known as the "Kawempe Community Health" study which ran from 2000 to 2012 [40]. We whole-genome sequenced 185 isolates, all belonging to *Mtb* sublineage 4.6.1 (raw reads available in the European Nucleotide Archive under project accession numbers PRJEB11460, PRJNA354716, and PRJEB64921). These isolates were obtained from HIV-positive and HIV-negative TB patients defined mostly as index cases within a household (with the exception of 4 and 3 strains from contact and co-prevalent cases, respectively). HIV status was determined by ELISA and confirmed by Western blot at baseline. CD4+ T-cell counts were available for 50

HIV-positive patients. The incidence rate of sputum smear-positive TB in Kampala in 2001–2002 was estimated around 370 cases per 100,000 people per year [39]. As only the year of sample isolation was available, all isolates were assumed to be collected on the 1st of January of the corresponding year.

## Whole-genome sequencing

Bacterial isolates from Kampala, Uganda were cultured on Middlebrook 7H10 supplemented with 10% glycerol and OADC until confluent colonies appeared on the plates. The colonies were scraped off and their genomic DNA was extracted using the CTAB method [61]. Selected strains were whole-genome sequenced on an Illumina HiSeq 2000 instrument at the commercial facility GATC (Germany). Library preparation was performed according to Illumina's TruSeq DNA Sample Preparation Guide (Illumina, San Diego, CA). Single-end sequence reads of approximately 50 bp were obtained. Demultiplexing was performed automatically by the CASAVA pipeline v1.8.0 (Illumina, San Diego, CA).

## WGS analyses and alignments

For all datasets, the Illumina reads were processed and analyzed as described in [37, 38]. Lineages and sublineages were identified using the SNP-based classification by Steiner et al. [62]. For all sequences per lineage and per location, an alignment of polymorphic positions was assembled by concatenating all high-quality SNPs. Sites that had more than 10% of missing data, as well as drug-resistance-related sites, were excluded from the alignment.

## Phylodynamic analyses

We fit a multitype birth-death model to the sequence alignments [43], with two types corresponding to HIV-negative TB cases and HIV-positive TB cases (S1 Fig). Under this model, a 'birth' event corresponds to an *Mtb* transmission event from one host to another, which can occur within and between types. A 'death' event occurs when a host becomes uninfectious for *Mtb* due to recovery or death. The model was parametrized with the $R_e$ within a purely HIV-negative population ($R_e^b$), the rate at which HIV-negative individuals become uninfectious ($\delta^-$), the multiplicative effect of HIV co-infection on transmitting *Mtb* ($f_1$), the multiplicative effect of HIV co-infection on the risk of TB disease development upon contact with an infectious individual ($f_2$), and the multiplicative effect of HIV co-infection on the rate of becoming uninfectious for *Mtb* ($f_3$). The effective reproductive number for TB within the HIV-negative subpopulation, within the HIV-positive subpopulation, from the HIV-negative to the HIV-positive subpopulation, and from the HIV-positive to the HIV-negative subpopulation ($R_e^{--}$, $R_e^{++}$, $R_e^{-+}$, and $R_e^{+-}$, respectively), as well as the rate at which HIV-positive individuals become *Mtb* uninfectious ($\delta^+$), are then as follows:

$$
\begin{aligned}
R_e^{--} &= (1 - p_{\text{HIV}})R_e^b \\
R_e^{++} &= f_1 f_2 p_{\text{HIV}} R_e^b \\
R_e^{-+} &= f_2 p_{\text{HIV}} R_e^b \\
R_e^{+-} &= f_1 (1 - p_{\text{HIV}})R_e^b \\
\delta^+ &= f_3 \delta^-
\end{aligned}
$$

$p_{\text{HIV}}$ represents the overall prevalence of HIV in the general population (including both *Mtb*-infected and *Mtb*-uninfected individuals) and is included to account for different sizes of the HIV-negative and HIV-positive populations. As the HIV prevalence in a country changed

over time since the date of HIV introduction, we let $p_{\text{HIV}}$ change at three different time points in the past, according to HIV prevalence data from World Bank [44–47] (S2 Fig). Hence, the effective reproductive numbers also changed through time.

The overall reproductive numbers for HIV-negative and HIV-positive individuals are

$$
\begin{aligned}
R_e^- &= R_e^{--} + R_e^{-+} = (1 - p_{\text{HIV}})R_e^b + f_2 p_{\text{HIV}} R_e^b \\
R_e^+ &= R_e^{+-} + R_e^{++} = f_1(1 - p_{\text{HIV}})R_e^b + f_1 f_2 p_{\text{HIV}} R_e^b
\end{aligned}
$$

From these equations, it can be seen that $f_1 = R_e^+/R_e^-$.

Transmission rates (i.e., rates of *Mtb* transmission per infected individual per unit of time) from HIV-negative and HIV-positive individuals can be obtained using the definition of the reproductive number:

$$
\begin{aligned}
\lambda^- &= R_e^- \delta^- \\
\lambda^+ &= R_e^+ \delta^+
\end{aligned}
$$

We assumed no migration between subpopulations, implying that HIV-negative individuals cannot get infected with HIV during their period of *Mtb* infectiousness. TB patients are sampled with sampling proportion *s*, which was set to zero before the onset of sampling. Upon sampling an infected patient, the patient is assumed to become uninfectious with probability *r* [63].

We further assumed a strict molecular clock and a general time-reversible nucleotide substitution model with four gamma rate categories to account for site-to-site rate heterogeneity (GTR+$\Gamma_4$).

We performed phylodynamic inference using the bdmm package [43] v1.0.3 (https://github.com/tgvaughan/bdmm/releases/tag/v1.0.3-unofficial), feast package v8.3.1 (https://github.com/tgvaughan/feast/releases/tag/v8.3.1), and skylinetools package v0.2.0 (https://github.com/laduplessis/skylinetools/releases/tag/0.2.0) in BEAST v2.6.6 [49, 50]. Data from each location were analyzed independently. For each location, variable SNP alignments were generated per *Mtb* lineage and augmented with a count of invariant A, C, G, and T nucleotides to avoid ascertainment bias [64]. To avoid unreasonably long runtimes, any alignment containing more than 400 sequences was randomly downsampled to 400 sequences, and sampling proportion priors were adjusted accordingly. Population dynamic parameters were inferred jointly for the different *Mtb* lineages within one location: each lineage was represented with an independent tree with its own origin time and nucleotide substitution parameters, but sharing all other parameters with the other lineages.

Three independent Markov Chain Monte Carlo chains were run for each analysis, with states sampled every 1,000 steps. Tracer [65] was used to assess convergence and confirm that the effective sample size (ESS) was at least 200 for the parameters of interest. 10% of each chain was discarded as burn-in, and the remaining samples across the three chains were pooled using LogCombiner [50], resulting in at least 300,000,000 iterations in combined chains.

## Prior distributions

All parameters and their corresponding prior distributions are listed in S2 Table. For the sampling proportion, a uniform prior was chosen with lower bound set to zero and upper bound set equal to the ratio of the number of sequences, corrected for downsampling, and the total number of reported cases during the sampling period (S4 Table).

## Sensitivity analyses

The robustness of the phylodynamic inference was assessed by changing the fixed value of $\delta^-$ to 0.5 and 2 year$^{-1}$, by changing the prior on $f_1$ and $f_2$ to a Lognormal(0,0.5) distribution, and by fixing the clock rate to $10^{-8}$ and $10^{-7}$ substitutions per site per year.

For the analyses with local HIV prevalence estimates, we set the local prevalence in Karonga (Malawi) to 0%, 6%, 13%, and 11% before 1986, between 1986 and 1993, between 1993 and 2005, and after 2005, respectively, based on estimates reported in [66] and [67]. For Khayelitsha (South Africa), the local prevalence was set to 0%, 13%, and 23% before 1990, between 1990 and 2008, and after 2008, respectively. These estimates were based on the antenatal maternal HIV prevalence reported in [68], converted to a prevalence estimate in the general adult population by using the women's share of adults living with HIV [69]. For the other sampling locations, accurate local estimates could not be obtained.

The effect of the relative sampling proportion was assessed by assuming a twofold lower sampling proportion for either HIV-positive or HIV-negative TB cases compared to the default value.

To evaluate the relative impact of the sequence data on our parameter estimates, a phylodynamic analysis was performed using the same setup as the main analyses, but without any sequence data.

## Birth-death skyline analysis

To investigate whether $R_e^b$ can be assumed constant through time, we ran a birth-death skyline analysis on sequences from the most abundant lineage per sampling location [48]. No population structure is assumed in this model. Two time intervals were used to estimate potential changes in the overall $R_e$ over time, with the change point set at the estimated time of HIV introduction (S2 Fig). The overall rate of becoming *Mtb* uninfectious was assumed constant through time. For the sampling proportion, clock model and substitution model parameters, the same settings and priors were used as in the multitype birth-death model.

## Randomization of HIV status

The HIV infection status of the patients was randomized in two ways. First, the HIV status labels were permuted, implying that the HIV prevalence among the sampled TB patients was kept unchanged. Second, the HIV status was randomly assigned to each patient, with an overall HIV prevalence among the patients assumed equal to the average HIV prevalence in the general population (including both *Mtb*-uninfected and *Mtb*-infected individuals) during the sampling period (S2 Fig) [44–47]. Both randomization procedures were replicated 10 times.

## Population stratification by CD4+ T-cell count

The TB patient population was stratified based on CD4+ T-cell count, with a threshold set at 350 cells/$\mu$l, corresponding to the threshold recommended by WHO to prioritize patients for ART (S15 Fig) [70]. As CD4+ T-cell counts were not monitored for HIV-negative TB patients from Uganda, these patients were all classified as having a high CD4+ T-cell count (in accordance with data from South Africa, S15 Fig). The fitted phylodynamic model was equivalent to the model based on HIV status, with HIV-negative individuals being replaced by individuals with a high CD4+ T-cell count ($\geq$ 350 cells/$\mu$l) and HIV-positive individuals being replaced by individuals with a low CD4+ T-cell count ($<$ 350 cells/$\mu$l). Correspondingly, $p_{\text{HIV}}$ was replaced by $p_{\text{lowCD4+}}$, the prevalence of cases with low CD4+ T-cell counts. This prevalence was estimated as follows: the HIV prevalence in the general population was multiplied by the observed

proportion of patients with low CD4+ T-cell count among HIV-positive patients in the dataset. $p_{\text{lowCD4+}}$ was set to 75% of this value, as HIV-positive patients with low CD4+ T-cell counts are likely overrepresented among TB patients. Changing this 75% to higher or lower values did not alter the qualitative conclusions.

## Statistical analyses

Associations between HIV infection status and other variables were tested using Welch's $t$-tests and $\chi^2$-tests implemented in R.

## Supporting information

**S1 Fig. Schematic representation of the phylodynamic model.** Phylodynamic model used to estimate HIV effects on *Mtb* transmission, based on a structured birth-death model with HIV-negative and HIV-positive TB cases representing different subpopulations. a) Each subpopulation has its own rate of becoming uninfectious (indicated as $\delta^-$ and $\delta^+$) and sampling rate (indicated as $s^-$ and $s^+$). Transmission events occur within each subpopulation with reproductive numbers indicated as $R_e^{--}$ and $R_e^{++}$, and between subpopulations with effective reproductive numbers indicated as $R_e^{-+}$ and $R_e^{+-}$. b) For the analyses in this study, the model was reparametrized by expressing the reproductive numbers as a function of a base $R_e$ ($R_e^b$), the HIV prevalence in the general population ($p_{\text{HIV}}$), the multiplicative effect of HIV co-infection on the $R_e$ of TB patients ($f_1$), the multiplicative effect of HIV co-infection on the risk of developing active TB when exposed ($f_2$), and the multiplicative effect of HIV co-infection on the rate of becoming uninfectious ($f_3$).
(TIF)

**S2 Fig. HIV prevalence in different countries over time.** Coloured lines represent the prevalence (% of population ages 15-49) per country over time, as reported by World Bank [44–47], while the dashed grey lines represent the values used in our phylodynamic model.
(TIF)

**S3 Fig. $R_e$ estimates resulting from the birth-death skyline analyses.** Prior (grey) and posterior (coloured) distributions per sampling location of the estimates of the overall $R_e$ before and after the estimated time of HIV introduction into the country, assuming no structure in the population.
(TIF)

**S4 Fig. Posterior maximum clade credibility trees of *Mtb* isolates from Malawi.** Posterior maximum clade credibility tree per lineage, summarizing the posterior tree distribution resulting from the phylodynamic analyses on the sequences from Malawi, with tips labeled by HIV infection status.
(TIF)

**S5 Fig. Posterior maximum clade credibility trees of *Mtb* isolates from South Africa.** Posterior maximum clade credibility tree per lineage, summarizing the posterior tree distribution resulting from the phylodynamic analyses on the sequences from South Africa, with tips labeled by HIV infection status.
(TIF)

**S6 Fig. Posterior maximum clade credibility tree of *Mtb* isolates from Uganda.** Posterior maximum clade credibility tree, summarizing the posterior tree distribution resulting from the phylodynamic analyses on the sequences from Uganda (lineage 4 only), with tips labeled

by HIV infection status.
(TIF)

**S7 Fig. Additional parameter estimates resulting from the main phylodynamic analyses.**
Posterior distributions per sampling location of the estimates for a) the relative *Mtb* infectious
period (HIV-positive relative to HIV-negative individuals), and b) the relative *Mtb* transmission rate. For all posterior distributions in (b), the 95% HPD intervals do not contain 1.
(TIF)

**S8 Fig. Parameter estimates for the sensitivity analyses on the becoming uninfectious rate.**
Prior (grey) and posterior (coloured) distributions per sampling location of the estimates for
a) the relative risk of developing active TB upon exposure (HIV-positive relative to HIV-negative individuals), assuming a fixed becoming uninfectious rate of 0.5 year$^{-1}$, b) the relative $R_e$
for TB, assuming a fixed becoming uninfectious rate of 0.5 year$^{-1}$, c) the relative risk of developing active TB upon exposure, assuming a fixed becoming uninfectious rate of 2 year$^{-1}$, and
d) the relative $R_e$ for TB, assuming a fixed becoming uninfectious rate of 2 year$^{-1}$. For all posterior distributions, the 95% HPD intervals do not contain 1.
(TIF)

**S9 Fig. Parameter estimates for the sensitivity analyses on the HIV effect priors.** Prior
(grey) and posterior (coloured) distributions per sampling location of the estimates for a) the
relative risk of developing active TB upon exposure (HIV-positive relative to HIV-negative
individuals), and b) the relative $R_e$ for TB, assuming Lognormal(0,0.5) priors on the effect of
HIV on *Mtb* transmission ($f_1$) and TB disease progression ($f_2$). For all posterior distributions,
the 95% HPD intervals do not contain 1.
(TIF)

**S10 Fig. Parameter estimates for the sensitivity analyses on the clock rate.** Prior (grey) and
posterior (coloured) distributions per sampling location of the estimates for a) the relative risk
of developing active TB upon exposure (HIV-positive relative to HIV-negative individuals),
assuming a fixed clock rate of $10^{-8}$ substitutions per site per year, b) the relative $R_e$ for TB,
assuming a fixed clock rate of $10^{-8}$ substitutions per site per year, c) the relative risk of developing active TB upon exposure, assuming a fixed clock rate of $10^{-7}$ substitutions per site per
year, and d) the relative $R_e$ for TB, assuming a fixed clock rate of $10^{-7}$ substitutions per site per
year. For all posterior distributions, the 95% HPD intervals do not contain 1.
(TIF)

**S11 Fig. Parameter estimates for the sensitivity analyses on the HIV prevalence.** Prior
(grey) and posterior (coloured) distributions per sampling location of the estimates for a) the
relative risk of developing active TB upon exposure (HIV-positive relative to HIV-negative
individuals), and b) the relative $R_e$ for TB, using local estimates of the HIV prevalence in Karonga (Malawi) and Khayelitsha (South Africa) (see Materials and methods). For all posterior
distributions, the 95% HPD intervals do not contain 1.
(TIF)

**S12 Fig. Parameter estimates for the sensitivity analyses on the relative sampling proportion.** Prior (grey) and posterior (coloured) distributions per sampling location of the estimates
for a) the relative risk of developing active TB upon exposure (HIV-positive relative to HIV-negative individuals), assuming a twofold undersampling of HIV-positive compared to HIV-negative cases, b) the relative $R_e$ for TB, assuming a twofold undersampling of HIV-positive
compared to HIV-negative cases, c) the relative risk of developing active TB upon exposure,
assuming a twofold oversampling of HIV-positive compared to HIV-negative cases, and d) the

relative $R_e$ for TB, assuming a twofold oversampling of HIV-positive compared to HIV-negative cases. For all posterior distributions, the 95% HPD intervals do not contain 1.
(TIF)

**S13 Fig. Parameter estimates for the analyses with and without genomic data.** a) Posterior distributions per sampling location of the estimates for the relative risk of developing active TB upon exposure (HIV-positive relative to HIV-negative individuals), only based on the sampling dates and HIV infection status (light colours), or also including the sequences (dark colours). b) Posterior distributions per sampling location of the estimates for the relative $R_e$ for TB, only based on the sampling dates and HIV infection status (light colours), or also including the sequences (dark colours).
(TIF)

**S14 Fig. Parameter estimates for the analyses on randomized datasets.** Prior (grey) and posterior (coloured) distributions per sampling location of the estimates for a) the relative risk of developing active TB upon exposure (HIV-positive relative to HIV-negative individuals), on 10 different datasets where the HIV status labels of the patients were permuted, b) the relative $R_e$ for TB, on 10 different datasets where the HIV status labels of the patients were permuted, c) the relative risk of developing active TB upon exposure, on 10 different datasets where the HIV status labels were randomly assigned using the average HIV frequency in the general population during the sampling period, d) the relative $R_e$ for TB, on 10 different datasets where the HIV status labels were randomly assigned using the average HIV frequency in the general population during the sampling period.
(TIF)

**S15 Fig. CD4+ T-cell counts of HIV-negative and HIV-positive TB patients from South Africa ($n$ = 1, 110) and Uganda ($n$ = 50).** In Uganda, CD4+ T-cell counts were only recorded for HIV-positive patients. The dashed line represents the threshold (350 cells/$\mu$l) recommended by WHO to prioritize patients for ART [70]. This threshold was used to stratify the TB patient population.
(TIF)

**S1 Table. Observed lineage distribution at the different sampling locations, based on the number of sequences in the datasets.**
(PDF)

**S2 Table. Prior distributions for the parameters of the multitype birth-death model.**
(PDF)

**S3 Table. Clock rate estimates resulting from the main phylodynamic analyses.**
(PDF)

**S4 Table. Total number of reported cases during the sampling period and total number of sequences included in the analyses at the different sampling locations.**
(PDF)

**S5 Table. Accession numbers and metadata associated with the *Mtb* genomes used in this study.**
(CSV)

## Acknowledgments

Calculations were performed on the Euler cluster at ETH Zürich and at sciCORE (https://scicore.unibas.ch/) scientific computing core facility at the University of Basel. We would like to thank Louis du Plessis for valuable feedback on the manuscript, Timothy G. Vaughan for help with the phylodynamic analyses, Cecilia Valenzuela Agüí and Ailene MacPherson for insightful discussions, and the Malawi Epidemiology and Intervention Research Unit for providing patient data.

## Author Contributions

**Conceptualization:** Etthel M. Windels, Sebastien Gagneux, Daniela Brites, Tanja Stadler.

**Data curation:** Etthel M. Windels, Daniela Brites.

**Formal analysis:** Etthel M. Windels, Galo A. Goig, Daniela Brites.

**Funding acquisition:** Moses L. Joloba, W. Henry Boom, Helen Cox, Sebastien Gagneux, Tanja Stadler.

**Investigation:** Etthel M. Windels.

**Methodology:** Etthel M. Windels, Tanja Stadler.

**Project administration:** Etthel M. Windels, Sonia Borrell.

**Resources:** Eddie M. Wampande, Moses L. Joloba, W. Henry Boom, Helen Cox, Jerry Hella.

**Supervision:** Daniela Brites.

**Visualization:** Etthel M. Windels.

**Writing – original draft:** Etthel M. Windels, Sebastien Gagneux, Daniela Brites, Tanja Stadler.

**Writing – review & editing:** Etthel M. Windels, W. Henry Boom, Galo A. Goig, Sonia Borrell, Sebastien Gagneux, Daniela Brites, Tanja Stadler.

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
