## [Decision Letter · Decision Letter 0]

4 Dec 2023

Dear Dr Windels,

Thank you very much for submitting your manuscript "HIV co-infection is associated with reduced *Mycobacterium tuberculosis* transmissibility in sub-Saharan Africa" for consideration at PLOS Pathogens. As with all papers reviewed by the journal, your manuscript was reviewed by members of the editorial board and by several independent reviewers. In light of the reviews (below this email), we would like to invite the resubmission of a significantly-revised version that takes into account the reviewers' comments.

The reviewers had some concerns regarding assumptions made in the modelling approach. Would the outcomes be the same is the model is run without dates? Can the reproductive number be assessed in patients with low CD4+ counts versus high CD4+ counts (the latter irrespective of HIV status)? The analyses need to be run based on HIV prevalence estimates amongst adults for each geographic region. The analyses need to include effect of initiation of ART.

We cannot make any decision about publication until we have seen the revised manuscript and your response to the reviewers' comments. Your revised manuscript is also likely to be sent to reviewers for further evaluation.

Sincerely,

Helena Ingrid Boshoff

Academic Editor

PLOS Pathogens

Michael Otto

Section Editor

PLOS Pathogens

Kasturi Haldar

Editor-in-Chief

PLOS Pathogens

orcid.org/0000-0001-5065-158X

Michael Malim

Editor-in-Chief

PLOS Pathogens

orcid.org/0000-0002-7699-2064

The reviewers had some concerns regarding assumptions made in the modelling approach. Would the outcomes be the same is the model is run without dates? Can the reproductive number be assessed in patients with low CD4+ counts versus high CD4+ counts (the latter irrespective of HIV status)? The analyses need to be run based on HIV prevalence estimates amongst adults for each geographic region. The analyses need to include effect of initiation of ART.

Reviewer's Responses to Questions

**Part I - Summary**

Reviewer #1: The manuscript uses phylodynamic data from four sub-Saharan African countries to estimate the relative infectiousness of Mtb in people living with HIV (PLHIV) with TB compared to HIV- people with TB. They show that 1) PLHIV are at higher risk of TB disease, and that 2) HIV coinfection is associated with a lower effective reproduction number. These findings are in line with previous studies. They also 3) attempt to determine the extent to which reductions in the reproduction number are due to lower durations of infectious vs lower transmission rates.

There is already a very strong evidence base for 1), with numerous studies providing estimates of RR generated using more direct and reliable methods, and systematic reviews summarising available evidence on risk by CD4 count (https://www.ncbi.nlm.nih.gov/pmc/articles/PMC5733368/) and ART (https://bmcinfectdis.biomedcentral.com/articles/10.1186/s12879-023-08533-0) This study therefore contributes little to scientific knowledge with 1).

There is a much weaker evidence base for 2, and therefore this study has the potential to make more valuable contributions to the literature. I have a number of concerns however, detailed below.

I find 3), the attempt to distinguish between infectious duration and relative transmission risk, very interesting, and reliable estimates of the contributions of the two factors would be valuable. I do not understand what it is in the model/data that allows the two to be distinguished however, and so cannot judge whether the results are likely to be reliable. I would greatly appreciate more discussion of the data and assumptions that go into distinguishing the two. (Also, see below for concerns around the South Africa estimates).

Reviewer #2: This manuscript models the effective reproductive number of TB among HIV and non-HIV risk groups in four different Sub-saharan african countries using a phylodynamic approach. There are many assumptions made to allow for this modeling with the advantages of integrating epidmiological, and genetic evolutionary parameters including a strict molecular clock, assumptions around population size changes over time, and the observed sampling dates of cases that were sequenced. This approach doesnot take into account individual or host risk factors, social network differences between people living HIV and those without HIV with regards to TB risk. The study also doesnot consider sampling differences for the patients and their samples that underwent whole genome sequences in the four different countries. One of the studies that they relied on Mtb WGS from South AFrica only sequenced drug resistant isolates. I believe that the authors have done an overall good job of listing limitations and assumptions of their approach, and I do believe they make a valuable contribution to the literature on the topic. I have a few comments that I hoped they could address:

**Part II – Major Issues: Key Experiments Required for Acceptance**

Reviewer #1: The genomic data came from specific study locations, however country-level HIV prevalence estimates were used. HIV prevalence can vary greatly by region, but also within regions. For instance, HIV prevalence is fairly low in Western Cape compared to South Africa as a whole, however it is likely to be higher in Khayelitsha than in the province as a while. Estimates for the study community should be available for Karonga at least, and potentially for Khayelitsha. The analyses should be repeated using the best available local (and adult only) HIV prevalence estimates.

ART reduces the risk of developing TB, and it is highly probable that it also increases TB infectiousness. Country-level ART coverage varied from 14% to 70% in South Africa, from 0% to 34% in Uganda, 0% to 35% in Malawi, and 35% to 75% in Tanzania over the course of the periods in which the genomic data were collected. This is not incorporated into the analysis approach (e.g. by allowing parameter values to vary over time), and it is therefore unclear what the estimates obtained represent, or if they can be generalized to present day settings with higher ART coverage.

The prior distributions around the relative progression risk for PLHIV are unrealistic – it certainty shouldn’t include numbers below 1, and there are plenty of data sources to provide a more reasonable prior. There is also little to support a prior on the relative R0 that goes above 1.

The posterior distribution for South Africa for the relative infectious period of HIV+ TB falls almost entirely above 1, and has a median of around 3. This lacks face validity – prevalence to notification ratios are 53% lower for PLHIV than for HIV- people (doi:10.1111/tmi.13485). It is also in contrast to the results for the other three settings.

Table S4 shows that only (very) small proportion of people with notified TB had samples included in the analysis. There will be biases in who made it into the dataset, which will vary between settings. This includes (but is not limited to) biases driven by study design – e.g. people diagnosed in hospital only in Tanzania. The large proportion of people with TB who are never diagnosed will also not be represented in the samples, and treatment coverage is likely to vary by HIV status. The implications of these biases on the results should be discussed.

Only sequences from patients with known HIV status were retained. What proportion of all sequences was this? What would be the potential impact of the probable differential missingness by HIV status? I did not understand the ‘Randomization of HIV status’ sensitivity analysis at all, I am not sure if that was designed to address this?

Reviewer #2: 1- Can the authors describe the sampling of WGS in the results in more detail? were the patients with TB all comers to clinic? were all patients recruited in those studies sequenced? or were some missing. Was the expected missing rate of culture and sequencing preferrentially higher for HIV (that is what I expect given patients with HIV are less likely to be microbiologically confirmed). What is the expected effect of undersampled patients with TB on their analysis results?

2- The authors describe that the date information was largely sufficient (over genomic and date data) to infer a lower transmission rate for HIV than for non-HIV index cases. The authors attribute this to dispersal of HIV/TB cases over the sampled TB phylogeny. Perhaps a related explanation is that the date and genetic data is highly correlated, can the authors run the model without dates and assess if the conclusions hold? related to this is whether there is sufficient temporal signal in the trees observed, and if they can elaborate on if molecular clock rates are being estimated or used from the literature. If they are being estimated how does the tree topology change with and without the dates?

3- THe authors should temper down the conclusions they draw about the lack of association between CD4 count and transmissibility, especially given the lack of ART data. Can they also clarify the total sample size that supported the CD4 analysis? Also can they specifically assess the reproductive number among index cases with HIV and low CD4 count vs as a group the (non-HIV patients + HIV with high CD4)?

**Part III – Minor Issues: Editorial and Data Presentation Modifications**

Reviewer #1: “The TB notification rate in Temeke in 2020 was 3,994 cases per year” – that is an absolute number of cases not a rate, and hard to interpret without knowing the population size

“ART only recently became widely accessible to HIV patients in South Africa and Uganda” – this is not true, e.g. estimated ART coverage in South Africa reached 50% in 2014.

“As we could only investigate the overall risk of developing active TB 188 disease after contact with a TB patient, it remains unclear, based on our data, whether 189 HIV also affects the risk of Mtb infection.” – it is the overall risk of developing TB, not risk after contact (if it is indeed the latter, then that needs to be made much clearer in the methods”)

Be careful about referring to people as ‘TB patients’ when you are talking about infectiousness – once they are diagnosed, become patients, and start treatment their infectiousness is greatly reduced – it is the period before they become patients where the vast majority of transmission is thought to occur

It should be made clear that the study is on pulmonary TB only

The link to the code doesn’t work

Reviewer #2: 4- minor comment, the second paragraph is not informative, I wouldn't descirbe the association between HIV and active TB as a "finding", and suggest deleting this paragraph.

PLOS authors have the option to publish the peer review history of their article (what does this mean?). If published, this will include your full peer review and any attached files.

Reviewer #1: No

Reviewer #2: **Yes: **Maha Farhat
---

## [Editor Report · Decision Letter 1]

10 Apr 2024

Dear Dr Windels,

We are pleased to inform you that your manuscript 'HIV co-infection is associated with reduced *Mycobacterium tuberculosis* transmissibility in sub-Saharan Africa' has been provisionally accepted for publication in PLOS Pathogens.

Best regards,

Helena Ingrid Boshoff

Section Editor

PLOS Pathogens

Michael Otto

Section Editor

PLOS Pathogens

Michael Malim

Editor-in-Chief

PLOS Pathogens

orcid.org/0000-0002-7699-2064

The authors have sufficiently addressed the reviewers' concerns.
---

## [Editor Report · Acceptance letter]

28 Apr 2024

Dear Dr Windels,

We are delighted to inform you that your manuscript, "HIV co-infection is associated with reduced *Mycobacterium tuberculosis* transmissibility in sub-Saharan Africa," has been formally accepted for publication in PLOS Pathogens.

Best regards,

Michael Malim

Editor-in-Chief

PLOS Pathogens

orcid.org/0000-0002-7699-2064